# Caregivers' use of herbal and conventional medicine to treat children with sickle cell disease at Jinja Regional Referral Hospital, Eastern Uganda: A cross-sectional study

**Consiliate Apolot**[1]*, **Samuel Baker Obakiro**[2,3], **David Mukunya**[1], **Peter Olupot-Olupot**[1,4], **Joseph K. B. Matovu**[1,5]

1 Department of Community and Public Health, Busitema University Faculty of Health Sciences, Mbale, Uganda, 2 Department of Pharmacology and Therapeutics, Busitema University Faculty of Health Sciences, Mbale, Uganda, 3 Natural Products Research and Innovation Centre, Busitema University Faculty of Health Sciences, Mbale, Uganda, 4 Mbale Clinical Research Institute, Mbale, Uganda, 5 Department of Disease Control and Environmental Health, Makerere University School of Public Health, Kampala, Uganda

* consiliate511@gmail.com

**Data Availability Statement:** All the data are within the data and its supporting information files.

## Abstract

### Background

Evidence suggests use of herbal and conventional medicines in the treatment of Sickle Cell Disease (SCD). We examined factors associated with caregivers' use of combined herbal and conventional medicine to treat children with SCD.

### Methods

A cross-sectional study was conducted at Jinja Regional Referral Hospital between January and March 2022. Caregivers of children with SCD aged 1 to 18 years attending the Sickle Cell Clinic were interviewed using structured questionnaires. We collected data on caregivers' socio-demographic characteristics, perceptions of and intentions to use either or both therapies, self-reported use of either or both therapies and community and health-related factors. A multivariable logistic regression model was computed to assess the factors independently associated with caregivers' use of combined therapy, using Stata version 15.0.

### Results

372 caregivers were interviewed. On average, respondents were aged 34.3 years (Standard Deviation [SD]: ±9.8 years). 37% (n = 138) of the caregivers reported the use of both herbal and conventional medicine, 58.3% (n = 217) reported use of only conventional medicine, while 4.6% (n = 17) reported use of herbal medicine only. Higher odds of using combination therapy were found in caregivers aged 60+ years (adjusted odds ratio [AOR] = 11.8; 95% CI: 1.2, 115.2), those with lower secondary education (AOR = 6.2; 95% CI: 1.5, 26.0), those who believed in the safety of herbal medicine (AOR = 3.3; 95% CI: 1.5, 7.6) and those who thought that use of both therapies were safe (AOR = 7.7; 95% CI: 3.5, 17.0).

**Funding:** The authors received no specific funding for this work.

**Competing interests:** The authors have declared that no competing interests exist.

## Conclusion

More than one-third of the caregivers reported use of combined herbal and conventional medicine, most of whom were older (>60%) and had lower secondary education. There is need for targeted health promotion to educate caregivers about the dangers of using both herbal and conventional medicines in treating children with SCD.

## Introduction

Globally, there is renewed interest in the use of complementary and alternative medicines (CAM) for disease management with a preponderance (80%) in Sub-Saharan Africa (SSA) [1, 2]. The use of herbal medicines varies from region to region ranging from 20% to 80% [1, 2]. Chronic health conditions such as sickle cell disease, diabetes, asthma, epilepsy, hypertension, HIV infection, and cancer [3–6] are among the common conditions in which complementary and alternative medicines are used. Worldwide, it is estimated that 300,000 babies are born annually with Sickle Cell Disease (SCD), most of whom (>90%) occur in SSA where about 10–40% of the population have the SCD trait [7]. The World Health Organization (WHO) approximates that SCD contributes to 6–16% of under-five mortality in many African countries, and this burden is projected to increase substantially in the next 40 years [8]. SCD also causes profound adverse health and socioeconomic effects among surviving children and their families, respectively [8]. In most parts of sub-Saharan Africa (SSA), SCD is a growing public health problem, but data on the burden of the disease is incomplete partly due to a lack of universal newborn screening, lack of SCD registries, and associated stigma. Within the current data limits, the vast part of Uganda has a sickle cell trait prevalence of between 9–21% with 1–2% of babies born with sickle cell disease (SCD) [7, 9]. These figures are high and rank this country the 5th with the highest burden of sickle cell disease in Africa [9].

In most SSA settings, care for SCD includes the prevention of bacterial and malaria infections through prophylactic medication. In addition, pneumococcal vaccinations, health education, and supplemental folic acid are given. Some SSA countries have introduced hydroxyurea, which is widely used in developed countries for the management of SCD. This novelty in these settings is gaining ground with new evidence of the benefits of its one among African populations [10, 11]. Given its chronic nature, treatment of SCD is often managed with other adjuvant therapies. Several studies have reported that there is growing use of herbal remedies in the management of symptoms in patients with SCD [12, 13]. This could be related to the perceived failure of conventional medicines in the community, amongst other notable factors which can be influenced by intrinsic or extrinsic factors.

This study was informed by Ajzen and Fishbein's theory of planned behavior (TPB), a widely studied social psychology model which determines the effect of consciously intended behaviors. The Theory of Planned Behaviour shows that behavioral intention, which predicts whether an individual will perform a behavior, can be predicted by attitudes, the subjective norm, and perceived behavioral control, which are seen as predictor variables. In the context of this study, behavior refers to the use of herbal medicine, conventional medicine, or both, and behavioral intention refers to the caregiver's determination to use either herbal medicine, conventional medicine, or both. Behavioral beliefs such as perceptions towards herbal and conventional medicine greatly influence the choice of treatment mode, as it directly impacts onto individual attitudes. Notably, literature has reported that some individuals have associated herbal medicine with perceived benefits, efficacy, and safety as compared to conventional medicines, where many dread drug side effects [13]

The historical use of herbal medicine pre-dates the industrial revolution, but formal descriptions of the concurrent use of herbal medicines and conventional medicines are a recent development. Among the medicinal plants commonly used for symptomatic management in SCD include aloe vera extracts, ginger, lemongrass, *Aframomum meleguet* (grains of paradise), garlic, *carica papaya* (unripe pawpaw), *sorghum bicolor* (leafstalk), seed of the *cajanus cajan* (pigeon peas) *piper guineensis* (dried fruit of ashanti pepper), *pterocarpus osun*, *eugenia caryophyllala* (cloves), and *sorghum bicolor*, *fagara* (f. zanthoxyloide) [12, 14]. The predictors favoring the utilization of herbal medicine, include affordability [13, 15], poor health service systems [6, 16], accessibility, and perceived efficacy. In addition, contributory factors comprise of shortage of conventional health professionals [17], inadequate health facilities, underfunding, mismanagement of the available health facilities [6], and cultural connotation [1, 2]. Remarkably, a study done in Sierra Leone reported that 14.8% of caregivers of adolescents with SCD faced catastrophic health expenditure during the course of treatment [18]. It is very likely that caregivers may end up oscillating between herbal and conventional medicine due to the financial strain that comes along with the chronicity of the disease. Furthermore, data establishes that the region of residency, being female, and living in homesteads with grandparents or elderly people contributed greatly to herbal medicine use [19].

Despite the growing use of CAM, concerns about the safety of using herbal medicine, along with conventional medicines, to treat children with SCD have been pointed out in the previous literature. For instance, some of these medicinal herbs may be intrinsically toxic especially since it was taken in non-standardized dosages [14]. Furthermore, concomitant use of herbal and conventional medicine may lead to counterproductive bidirectional drug-drug interactions. This is a great concern for sickle cell disease which is often associated with deteriorating multiorgan function including the kidneys, liver, and skeletal system, all of which participate in drug metabolic processes.

Previous studies have investigated the use of herbal medicine in sickle cell disease management [13]. However, to the best of our knowledge, no comprehensive studies have investigated the combined use of herbal and conventional medicine in sickle cell disease management using the theory of planned behavior. The study aimed to determine the prevalence and factors associated with caregivers' use of herbal medicine and conventional medicine in the management of children with SCD.

## Materials and methods

### Study design and site

This was a cross-sectional study conducted at Jinja Regional Referral Hospital (JRRH), a 600-bed hospital in Jinja City, Eastern Uganda. It serves eleven surrounding districts of Bugiri, Iganga, Jinja, Kaliro, Kamuli, Kayunga, Mayuge, Luuka, Namayingo, Namutumba, and Buyende with an estimated catchment population of 3.5 million people. Annexed to Jinja Hospital is Nalufenya Children's Hospital, from which the Sickle cell clinic operates [20]. The Sickle Cell Clinic (SCC) generally serves over 500 patients, and runs routinely, every Monday. During clinic days, i.e., on Mondays, the clinic serves about 30–50 patients. The choice for Jinja area was because it is ranked as one of the districts with a high burden for SCD in Uganda.

### Study population

This study was conducted among caregivers of children with sickle cell disease, enrolled in the Sickle Cell Clinic at Jinja Regional Referral Hospital. Caregivers were eligible for this study if they were caring for a child aged 1 to below 18 years. In this study's context, a caregiver was

referred to as any individual that is directly tending to the daily needs of the child living with sickle cell disease. For most caregivers of children with sickle cell disease, by the time the child is one year old, they have experienced the disease burden and are much aware of the chronicity and incurability of the disease and are therefore most likely to try other remedies outside conventional medicine. We excluded caregivers of children with sickle cell disease who were less than one year or had other chronic illnesses. This study considered a 62.9 prevalence of herbal use to manage sickle cell disease from a previous study by Busari et al. 2017 [13] with a precision of 5%, 95% confidence intervals, and a non-response of 10%. Based on these assumptions, we estimated a sample size of 394 participants.

## Sampling procedures

Official written consent to review registers of SCD clients was sought from the administration of JRRH. A retrospective review of patients' data was conducted to determine the number of sickle cell patients and their locations over 3 years, from January 2019 to January 2021. Following the retrospective review, we identified 607 children with SCD registered in care over the three-year period. Of these, 412 children were in the age bracket (1 to below18 years), and therefore eligible to participate in the study. The Sickle Cell Disease register comprised the children's date of last attendance at the SCC, child's name, name of Next of Kin (NOK), address of the NOK at the time of registration, contact details of the NOK, and child's age on the day of registration. A list of eligible children and their caregivers was generated. Caregivers were then identified through consecutive sampling. Phone calls were made to the eligible caregivers who were briefed about the study and scheduled for interviews. Caregivers whose phones were not reachable and those who reported that the children were already above 18 years were excluded. All participants eligible for the study provided written informed consent prior to the interview.

## Data collection procedures and methods

Data were collected by trained research assistants using a pretested structured interviewer-administered questionnaire developed based on the Theory of Planned Behaviour [21] and the literature review. The questionnaire was pretested on 5 caregivers, who were randomly chosen from the sickle cell clinic but who did not participate in the study. Pretesting was very helpful in ensuring that the questions were appropriate to the study population. The lead researcher (CA) worked with the research assistants to coordinate and mobilize the respondents. This was done to ensure that research standard operating procedures, quality, and data consistency were maintained throughout the study. The study's aim and rationale were explained to caregivers and only those who volunteered to participate took part. Data were collected on caregivers' socio-demographic characteristics, caregivers' experiences in disease management; and individual, community, and health-related factors that predicted the use of HM, CM or both therapies. Data were also collected on the theoretical constructs of the Theory of Planned Behavior, that is; attitudes towards use of HM, CM or both therapies, subjective norms about use of HM, CM or both therapies, intention to use HM or CM and caregivers' perceived behavioral control, defined as caregivers' perception of the ease or difficulty of using either HM, CM or both therapies [21].

## Data management and analysis

Questionnaires were checked for completeness by the lead researcher at the end of the data collection process and stored in locked cabinets that were accessible only by authorized study staff. Data were analyzed using Stata version 15.0. We presented descriptive statistics as frequencies and means. We conducted regression analysis for the combined use of herbal and

conventional medicine as the primary outcome. All caregivers who reported using both therapies were coded as one (1) while the rest (those who used either herbal or conventional medicine alone) were coded as zero (0). Logistic regression was used to compute the inferential statistics (i.e. determine the p-values and 95% confidence intervals) since the primary outcome was binary. All factors (that had a p<0.20 value at bivariate analysis were entered into the multivariate model. All factors that had a p-value of <0.05 in the final model were considered to be significantly associated with the primary outcome. Confounders and effect modifiers were controlled for statistically. All the possible confounders were included as control variables in the regression models. However, other confounding variables that might not have been accounted for at observation might still have remained.

### Ethical considerations

We obtained ethical approval from Mbale Regional Referral Hospital Research and Ethics Committee, approval number Mbale Regional Referral Hospital-2021-106. We further sought administrative clearance from the Hospital Director of Jinja Regional Referral Hospital and the Head of the Department of Paediatrics Section within the hospital. We obtained written informed consent from the participants before collecting data. Confidentiality and privacy of the study participants' information were maintained throughout the study. Participation in the study was voluntary; participants had a right to withdraw from the study at any time without necessarily giving a reason.

## Results

### Respondents' characteristics

Table 1 shows the characteristics of the caregivers interviewed as part of this study. A total of 372 (representing 94.4% of the sample) caregivers were interviewed during the study of whom 341 (91.7%) were females. The average age of respondents was 34.3 years (Standard Deviation [SD]: ±9.8 years). More than half (55.1%, n = 205) were aged 18 to 34 years, 88.4% (n = 329) were not married, 40.9% (n = 152) had primary education as their highest level of education, 29.6% (n = 110) were Protestants while 51.3% (n = 191) were Basoga by tribe.

### Use of herbal medicine, conventional medicine, and both herbal medicine and conventional medicine

Fig 1 shows the prevalence of use of herbal, conventional and combined therapies by caregivers of children with SCD. Overall, more than half of the caregivers (58.3%, n = 217) used only conventional medicine, slightly more than a third (37.1%, n = 138) used both conventional medicine and herbal medicine while 4.6% (n = 17) used herbal medicine alone.

Table 2 shows the distribution of the use of herbal medicine, conventional medicine, and both herbal medicine and conventional medicine among children with sickle cell disease by caregiver characteristics. Use of herbal medicine alone was common among men (12.9%, n = 4), caregivers aged 35–59 years (7%, n = 11), Muslims (7.4%, n = 8), those with upper secondary education (14.3%, n = 2) and subsistence farmers (7.8%, n = 8). Use of conventional medicine alone was common among females (59.2%, n = 202); caregivers aged 18–34 years (69.3%, n = 142), and those with no formal education (81.4%, n = 35). Use of combined herbal and conventional medicine was common among caregivers aged 60+ years (77.8%, n = 7), Muslims (43.5%, n = 47), Pentecostal/Born-again Christians (42.9%, n = 24), caregivers with tertiary/University education (50%, n = 12) and those in formal employment or professional jobs (54.1%, n = 20).

**Table 1. Socio-demographic characteristics of the caregivers of children with sickle cell disease at Jinja Regional Referral Hospital.**

| Characteristics | Frequency (n, %) | Percentage(%) |
|---|---|---|
| **Sex** | | |
| Female | 341 | 91.7 |
| Male | 31 | 8.3 |
| **Age-group** | | |
| 18–34 | 205 | 55.1 |
| 35–59 | 158 | 42.5 |
| 60+ | 9 | 2.4 |
| **Marital status** | | |
| Not married | 329 | 88.4 |
| Married | 43 | 11.6 |
| **Religion** | | |
| Catholic | 85 | 22.8 |
| Muslim | 108 | 29.0 |
| None | 13 | 3.5 |
| Pentecostal/Born again | 56 | 15.1 |
| Protestant /Anglican | 110 | 29.6 |
| **Ethnicity** | | |
| Baganda | 84 | 22.6 |
| Banyoli | 28 | 7.5 |
| Basoga | 191 | 51.3 |
| Other | 69 | 18.5 |
| **Highest level of education** | | |
| No formal education | 43 | 11.6 |
| Primary | 152 | 40.9 |
| Lower Secondary | 139 | 37.4 |
| Upper secondary | 14 | 3.8 |
| Tertiary university | 24 | 6.5 |
| **Occupation** | | |
| Casual workers for wages | 94 | 25.3 |
| Formal employment/professional | 25 | 6.7 |
| Subsistence farmer | 186 | 50.0 |
| Unemployed | 67 | 18.0 |

## Use of combined therapy by caregivers of children with sickle cell disease stratified by selected sickle cell disease management characteristics

Table 3 shows the use of combined therapy by caregivers of children with sickle cell disease stratified by selected sickle cell disease management characteristics. Use of combined therapy was higher among caregivers who reported than they used herbal medicine when they noticed that the child had sickle cell disease (43.3%, n = 13) than among those who started by administering conventional medicine at that time (36.5%, n = 125). About four in ten caregivers who had ever heard about hydroxyurea or were currently using hydroxyurea reported that they were using combined therapy. Use of combined therapy was higher among those who had used hydroxyurea for six or more months (50.0%, n = 13) than those who had used the drug for less than six months (30.8%, n = 8).

Interestingly, only 39.2% (n = 129) of those who had ever heard that herbal medicines are used for the management of SCD reported use of combined therapy but source of information

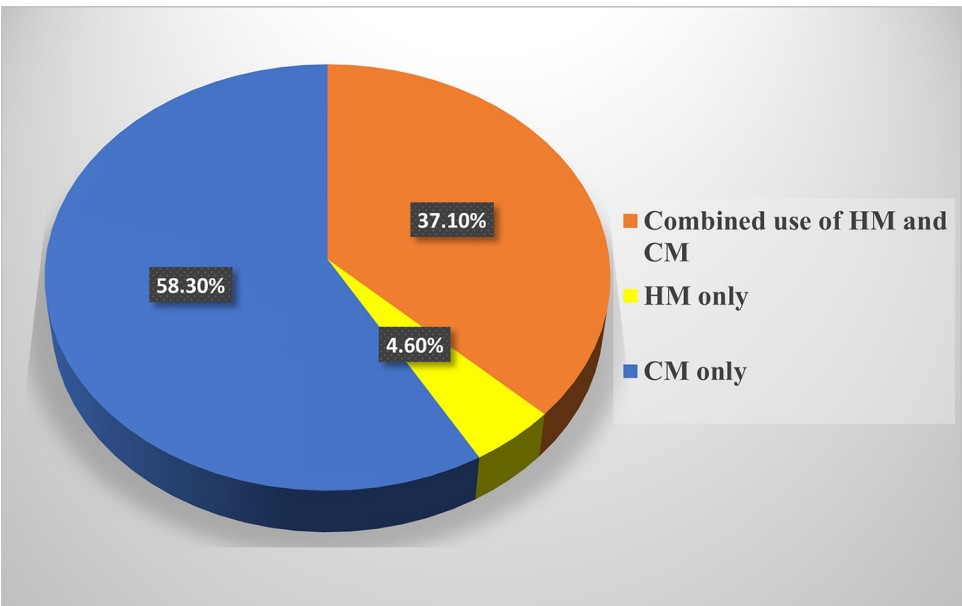

**Fig 1. Prevalence of use of herbal medicine, conventional medicine, and both herbal and conventional medicines by caregivers of children with SCD.**

about herbal medicine seemed to influence use of combined therapy ($P = 0.066$). A higher percentage of caregivers who heard about herbal medicine from neighbours/friends (50.8%, n = 32); those who heard this information from health professionals (50%, n = 4) and nearly half of those who heard this information from herbalists (45%, n = 18) used combined therapy than those who heard about herbal medicine from other sources. Interestingly, half of the caregivers who reported that they discussed herbal medicine usage with a physician reported that they used combined therapy.

### Caregivers' level of agreement with selected behavioral beliefs about the efficacy of herbal and conventional medicine and the use of combined herbal and conventional medicine

Table 4 shows the association between caregivers' level of agreement with selected behavioral beliefs about the efficacy of herbal medicine and the use of combined therapy among caregivers of children with sickle cell disease. As shown, caregivers who agreed with the statement that herbal medicine is more effective than conventional medicine and those who agreed that herbal medicine possesses lesser side effects than conventional medicine were significantly more likely to report that they used both herbal and conventional medicine in managing the disease than those who disagreed with these beliefs ($p < 0.05$). Similarly, caregivers who agreed with the statement that herbal medicine is cheaper than conventional medicine or that herbal medicine is usually used as a supplementary rather than alternative to conventional medicine were also relatively more likely to use a combined therapy than those who disagreed with those beliefs.

Table 5 shows the association between the caregivers' level of agreement with selected behavioral beliefs about the efficacy of conventional medicine and the use of combined therapy among caregivers of children with sickle cell disease. Caregivers who agreed with the statement that conventional medicine works better when used with herbal medicine, those who agreed that the high cost of conventional medicine made people resort to using herbal medicines, and

**Table 2. Use of herbal medicine, conventional medicine, and combined therapy by caregivers of children with Sickle Cell Disease by caregiver characteristics.**

| Characteristic | Total | HM only (n, %) | CM only (n, %) | Both HM and CM (n, %) |
|---|---|---|---|---|
| Overall | 372 | 17 (4.6) | 217 (58.3) | 138 (37.1) |
| **Sex** | | | | |
| Female | 341 | 13 (3.8) | 202 (59.2) | 126 (37.0) |
| Male | 31 | 4 (12.9) | 15 (48.4) | 12 (38.7) |
| **Age-group** | | | | |
| 18–34 | 205 | 6 (2.9) | 142 (69.3) | 57 (27.8) |
| 35–59 | 158 | 11 (7.0) | 73 (46.2) | 74 (46.8) |
| 60+ | 9 | 0 (0.0) | 2 (22.2) | 7 (77.8) |
| **Marital status** | | | | |
| Not married | 329 | 15 (4.6) | 192 (58.4) | 122 (37.1) |
| Married | 43 | 2 (4.7) | 25 (58.1) | 16 (37.2) |
| **Religion** | | | | |
| Atheist | 13 | 0 (0.0) | 9 (69.2) | 4 (30.08) |
| Catholic | 85 | 1 (1.2) | 58 (68.2) | 26 (30.6) |
| Muslim | 108 | 8 (7.4) | 53 (49.1) | 47 (43.5) |
| Pentecostal/Born again | 56 | 2 (3.6) | 30 (53.6) | 24 (42.9) |
| Protestant | 110 | 6 (5.5) | 67 (60.9) | 37 (33.6) |
| **Ethnicity** | | | | |
| Baganda | 84 | 2 (2.4) | 60 (71.4) | 22 (26.2) |
| Banyoli | 28 | 1 (3.6) | 18 (64.3) | 9 (32.1) |
| Basoga | 191 | 10 (5.2) | 98 (51.3) | 83 (43.5) |
| Other | 69 | 4 (5.8) | 41 (59.4) | 24 (34.8) |
| **Highest level of education** | | | | |
| No formal education | 43 | 1 (2.3) | 35 (81.4) | 7 (16.3) |
| Primary | 152 | 9 (5.9) | 90 (59.2) | 53 (34.9) |
| Lower Secondary | 139 | 5 (3.6) | 72 (51.8) | 62 (44.6) |
| Upper secondary | 14 | 2 (14.3) | 8 (57.1) | 4 (28.6) |
| Tertiary/university | 24 | 0 (0.0) | 12 (50.0) | 12 (50.0) |
| **Occupation** | | | | |
| Casual workers for wages | 129 | 6 (4.7) | 78 (60.5) | 45 (34.9) |
| Formal employment/ professional | 37 | 1 (2.7) | 16 (43.2) | 20 (54.1) |
| Subsistence farmer | 103 | 8 (7.8) | 63 (61.2) | 32 (31.1) |
| Unemployed | 103 | 2 (1.9) | 60 (58.3) | 41 (39.8) |

those who agreed that there was a need to integrate the use of conventional medicine with herbal medicine in the management of sickle cell disease were significantly more likely to report that they used both herbal medicine and conventional medicine than those who disagreed with these beliefs (p< 0.0001).

## Caregivers' use of combined herbal and conventional medicine by selected Theory of Planned Behavior constructs

Table 6 shows caregivers' use of combined herbal and conventional medicine by the selected Theory of Planned Behavior (TPB) constructs (i.e. intention, attitude, and subjective norms). Our findings show that intention to use both therapies and a good attitude towards the use of both, e.g. the belief that the use of both herbal and conventional medicine is both safe and beneficial, influenced the use of both herbal and conventional medicine. A higher proportion

**Table 3. Use of combined therapy by caregivers of children with sickle cell disease by selected sickle cell disease management characteristics.**

| Variable | Total | Use of both herbal and conventional medicines | | P-value |
|---|---|---|---|---|
| | | No (n = 350, %) | Yes (n = 225, %) | |
| **Kind of management given to the child on noticing the child had SCD** | | | | 0.461 |
| Conventional medicine | 342 | 217(63.4) | 125(36.5) | |
| Herbal medicine | 30 | 17(56.7) | 13(43.3) | |
| **Heard about hydroxyurea capsules** | 147 | 86(58.5)+ | 61(41.5) | 0.156 |
| **Child currently taking Hydroxyurea capsules** | 60 | 36(60.0) | 24(40.0) | 0.823 |
| **How long the child has been on hydroxyurea** | | | | 0.363 |
| Discontinued use | 8 | 5(62.5) | 3(37.5) | |
| Less than 6 months | 26 | 18(69.2) | 8(30.8) | |
| Over 6 months | 26 | 13(50.0) | 13(50.0) | |
| **Heard that herbal medicines are used for the management of SCD** | 329 | 200(60.8) | 129(39.2) | 0.0109 |
| **Source of information about herbal medicine** | | | | 0.066 |
| Fellow caregivers of children with sickle cell | 95 | 60(63.2) | 35(36.8) | |
| Health professionals | 8 | 4(50.0) | 4(50.0) | |
| Herbalists | 40 | 22(55.0) | 18(45.0) | |
| Media (TVs, radios, internet) | 93 | 67(72.0) | 26(28.0) | |
| Neighbors /friends | 63 | 31(49.2) | 32(50.8) | |
| Relatives | 30 | 16(53.3) | 14(46.7) | |
| **Ever discussed herbal medicine usage with the physician** | 34 | 17(50.0) | 17(50.0) | 0.103 |

(60.3%, n = 79) of caregivers who believed that the opinions of their friends regarding their choice of treatment for sickle cell disease management were important to them reported higher use of combined therapy than those who believed in the opinions of their family members (49.0%, n = 108) or the health professionals (37.5%, n = 132). However, only 38.1% (n = 118) of those who believed that the decision to choose the mode of treatment was under their complete control reported that they used combined therapy, with a slightly higher percentage of use of combined therapy reported among those who believed that the decision to use a given mode of treatment is dictated by its availability and accessibility (46.4%, n = 129) or by the dictates of one's surrounding (48.2%, n = 109).

## Factors associated with the use of combined therapy by caregivers of children with sickle cell disease

Table 7 shows the factors associated with the use of combined therapy by caregivers of children with sickle cell disease. Caregivers aged 60+ years were nearly 12 times more likely to use both herbal medicine and conventional medicine compared to those aged 18–35 years (adjusted odds ratio [AOR] = 11.8; 95% Confidence Interval [95%CI]: 1.2, 115.2). The use of both conventional medicine and herbal medicine was strongly associated with caregivers who had attained lower secondary education (AOR = 6.2; 95% CI: 1.5, 26.0) compared to those with no formal education. The caregivers who agreed that they intended to only use conventional medicine for disease management were 90% less likely to use both conventional medicine and herbal medicine (AOR = 0.1; 95% CI: 0.1, 0.3). There was notably a strong association between the likelihood of caregivers who agreed that herbal medicines are beneficial (AOR = 3.3; 95% CI: 1.5, 7.6) and those who agreed that the use of both herbal medicine and conventional medicine is safe (AOR = 7.7; 95% CI: 3.5, 17.0) with the use of both herbal and conventional medicine.

**Table 4. Association between caregivers' level of agreement with selected behavioral beliefs about the efficacy of herbal medicine and the use of combined therapy by caretakers of children with sickle cell disease.**

| Caregivers' level of agreement with behavioral beliefs about the efficacy of herbal medicine | Total N = 372 | Both HM and CM use | | OR(95% CI) | P-value |
|---|---|---|---|---|---|
| | | No (n, %) | Yes (n, %) | | |
| Herbal medicine cures symptoms faster than conventional medicine | | | | | 0.0001 |
| Disagree | 248 | 186(75.0) | 62(25.0) | 1.0 | |
| Agree | 124 | 86(69.4) | 38(30.6) | 4.9(3.1, 7.7) | |
| Herbal medicine has fewer side effects compared to conventional medicines | | | | | 0.0001 |
| Disagree | 216 | 176(81.5) | 40(18.5) | 1.0 | |
| Agree | 156 | 58(37.2) | **98(62.8)** | 7.4(4.6, 11.9) | |
| Herbal medicine is more effective in treating SCD than conventional medicine. | | | | | 0.0001 |
| Disagree | 269 | 199(74.0) | 70(26.0) | 1.0 | |
| Agree | 103 | 35(33.9) | **68(66.1)** | 5.5(3.4, 8.9) | |
| Herbal medicine is used as a supplementary therapy rather than an alternative to conventional medicine | | | | | 0.0001 |
| Disagree | 119 | 104(87.4) | 15(12.6) | 1.0 | |
| Agree | 253 | 130(51.4) | **123(48.6)** | 5.0(3.0, 8.5) | |
| People use herbal medicine for disease management because it is cheaper than conventional medicine | | | | | 0.0001 |
| Disagree | 177 | 133(75.1) | 44(24.9) | 1.0 | |
| Agree | 194 | 100(51.6) | 94(48.4) | 6.6(3.6, 11.9) | |
| People use herbal medicine for disease management because it is very easy to access information concerning the use of herbal medicine in managing sickle cell disease in our community | | | | | 0.0001 |
| Disagree | 84 | 60(71.4) | 24(28.6) | 1.0 | |
| Agree | 288 | 174(60.4) | 114(39.6) | 2.8(1.8, 4.4) | |
| People use herbal medicine because of cultural influence | | | | | 0.230 |
| Disagree | 103 | 58(56.3) | 45(43.7) | 1.0 | |
| Agree | 268 | 176(65.7) | 65(24.3) | 1.6(0.9, 2.8) | |
| Herbal medicine is used because herbal medicine is readily available in my community | | | | | 0.0001 |
| Disagree | 52 | 36(69.2) | 16(30.8) | 1.0 | |
| Agree | 319 | 197(61.8) | 122(38.2) | 2.7(1.6, 4.2) | |
| Attitudes of healthcare workers toward patients greatly influence the use of herbal medicine. | | | | | 0.0001 |
| Disagree | 177 | 123(69.5) | 54(30.5) | 1.0 | |
| Agree | 195 | 111(56.9) | 84(43.1) | 0.7(0.4, 1.1) | |
| Long distances to and from the health facility are what make people resort to using herbal medicine in treating children with SCD. | | | | | 0.436 |
| Disagree | 69 | 45(65.2) | 24(34.8) | 1.0 | |
| Agree | 303 | 189(62.4) | 114(37.6) | 1.4(0.7, 2.6) | |
| Good communication between vendors of herbal medicine and clients makes patients resort to using herbal medicines. | | | | | 0.002 |
| Disagree | 63 | 40(63.5) | 23(36.5) | 1.0 | |
| Agree | 309 | 194(62.8) | 115(37.2) | 2.0(1.3, 3.3) | |

## Discussion

Our findings showed that one-third (37.1%) of the caregivers used both herbal and conventional medicine for the treatment of their children with sickle cell disease. The relatively high prevalence of the use of both therapies may be attributed to the historical-cultural connotations Africans have attached to the use of herbal medicines [1, 2]. These trends, however, are not consistent throughout Africa, as findings from a study conducted in Ghana found a much lower prevalence of herbal use of 17.9% [22]. These two extremes suggest varying levels of

**Table 5. Association between caregivers' level of agreement with behavioral beliefs about the efficacy of conventional medicine and the use of combined therapy by caregivers of children with sickle cell disease.**

| Caregivers' level of agreement with behavioral beliefs about the efficacy of conventional medicine | Total | Both HM and CM use | | COR(95% CI) | P-value |
|---|---|---|---|---|---|
| | N = 372 | No (n, %) | Yes (n, %) | | |
| Conventional medicine works better when used with herbal medicine | | | | | |
| Disagree | 138 | 114(82.6) | 24(17.4) | 1.0 | |
| Agree | 236 | 120(50.8) | 116(49.2) | 5.0(3.0, 8.5) | 0.0001 |
| The high cost of conventional medicines makes people resort to using herbal medicines | | | | | |
| Disagree | 155 | 117(75.5) | 38(24.5) | 1.0 | |
| Agree | 216 | 116(53.7) | 100(46.3) | 2.7(1.7, 4.2) | 0.0001 |
| There is a need for the integration of herbal and conventional medicine in the management of sickle cell disease | | | | | |
| Disagree | 127 | 103(81.1) | 24(18.9) | 1.0 | |
| Agree | 245 | 131(53.5) | 114(46.5) | 3.7(2.2, 6.2) | 0.001 |

development of the combined use of herbal and conventional medicine. Despite the variations in the prevalence of use of either herbal medication alone or in combination with conventional medicines, we note the global nature of the practice as all different regions have a degree of use. This immense variation in prevalence in the different settings is probably associated with discrepant comprehension of the use of herbal medicine and conventional medicine by various researchers because herbal medicine has always been a fundamental part of the African healthcare system.

**Table 6. Caregivers' use of combined herbal and conventional medicine by selected Theory of Planned Behavior constructs.**

| Variable (Agree) | Total n = 372 | Both HM and CM use | | COR(95% CI) | P-value |
|---|---|---|---|---|---|
| | | No (n, %) | Yes (n, %) | | |
| **Intention(Ref = Disagree)** | | | | | |
| Intend to use only conventional medicine for the treatment of disease | 294 | 224 (76.2) | 70 (23.8) | 0.04(0.02, 0.1) | 0.001 |
| Intend to only use herbal medicines for the treatment of disease | 85 | 58 (68.2) | 27 (31.8) | 0.7(0.4, 1.2) | 0.001 |
| Intend to use both herbal and conventional medicines for the treatment of disease | 196 | 64 (32.7) | 132 (67.3) | 58.4(24.6, 139.1) | 0.001 |
| **Attitude(Ref = Disagree)** | | | | | |
| Herbal medicines are safe | 132 | 39 (29.5) | 93 (70.5) | 10.3(6.3, 16.9) | 0.001 |
| Herbal medicines are beneficial | 158 | 45 (28.5) | 113 (71.5) | 19.9(11.4, 34.6) | 0.001 |
| Conventional medicines are safe | 329 | 215 (65.3) | 114 (34.7) | 0.4(0.3, 0.8) | 0.006 |
| Conventional medicines are beneficial | 357 | 226 (63.3) | 131 (36.7) | 0.7(0.2, 1.9) | 0.436 |
| Use of both conventional medicines and herbal medicines are safe | 147 | 34 (23.1) | 113 (76.9) | 26.6(15.1, 46.8) | 0.001 |
| Use of both conventional medicines and herbal medicines are beneficial | 161 | 37 (23.0) | 124 (77.0) | 47.2(24.5, 90.8) | 0.001 |
| **Subjective norms(Ref = Disagree)** | | | | | |
| The opinion of your friends regarding your choice of treatment for sickle cell disease management is important to you | 131 | 52 (39.7) | 79 (60.3) | 4.7(3.0, 7.4) | 0.001 |
| The opinion of your family regarding your treatment choice is very important to you. | 220 | 112 (51.0) | 108 (49.0) | 3.9(2.4, 6.3) | 0.001 |
| The opinion of health professionals regarding your treatment choice is very important to you | 352 | 220 (62.5) | 132 (37.5) | 1.4(0.5, 3.7) | 0.501 |
| **Perceived behavioral control (Ref = Disagree)** | | | | | |
| You think that the decision to choose the mode of treatment is under your complete control | 310 | 192 (61.9) | 118 (38.1) | 1.3(0.7, 2.3) | 0.388 |
| The availability and accessibility of a treatment mode makes it an easy choice for caregivers to use | 278 | 149 (53.6) | 129 (46.4) | 8.2(4.0, 16.9) | 0.001 |
| Caregivers use either herbal or conventional medicines because their surrounding dictates it | 226 | 117 (51.8) | 109 (48.2) | 3.7(2.3, 6.0) | 0.001 |

**Table 7. Factors associated with the use of combined therapy by caregivers of children with sickle cell disease.**

| Variable | Total (N = 372) | Used combined therapy Yes (n, %) | Crude Odds Ratio (95% CI) | Adjusted Odds Ratio (95% CI) |
|---|---|---|---|---|
| **Age** | | | | |
| 18–34 | 205 | 57(41.3) | 1.0 | 1.0 |
| 35–59 | 158 | 74(53.6) | 2.3 (1.5, 3.5) | 1.3 (0.6, 2.7) |
| 60+ | 9 | 7(5.1) | 9.1 (1.8, 45.1) | 11.8 (1.2, 115.2)*** |
| **Ethnicity** | | | | |
| Baganda | 84 | 22(15.9) | 1.0 | 1.0 |
| Banyoli | 28 | 9(6.5) | 1.3 (0.5, 3.4) | 0.9 (0.2, 3.9) |
| Basoga | 191 | 83(60.1) | 2.2 (1.2, 3.8) | 1.4 (0.6, 3.6) |
| Other | 69 | 24(17.4) | 1.5 (0.8, 3.0) | 1.1 (0.3, 3.3) |
| **Highest level of education** | | | | |
| No formal education | 43 | 7(5.1) | 1.0 | 1.0 |
| Primary | 152 | 53(38.4) | 2.8 (1.1, 6.6) | 2.5 (0.7, 9.4) |
| Lower secondary (S1 to S4) | 139 | 62(44.9) | 4.1 (1.7, 9.9) | 6.2(1.5, 26.0)*** |
| Upper secondary (S5 to S6) | 14 | 4(2.9) | 2.1 (0.5, 8.5) | 4.0 (0.5, 32.0) |
| Tertiary university | 24 | 12(8.7) | 5.1 (1.6, 16.1) | 2.6 (0.4, 19.1) |
| **Occupation** | | | | |
| Casual workers for wages | 94 | 34(24.6) | 1.0 | 1.0 |
| Formal employment professional | 25 | 17(12.3) | 2.2 (1.0, 4.6) | 1.3 (0.3, 5.2) |
| Subsistence farmer | 186 | 61(44.2) | 0.8 (0.5, 1.5) | 1.7 (0.7, 4.2) |
| Unemployed | 67 | 26(18.8) | 1.2 (0.7, 2.1) | 1.0 (0.4, 2.6) |
| Heard that Herbal medicines are used for the management of SCD | 329 | 129(94.2) | 2.7 (1.2, 6.1) | 1.5 (0.5, 4.9) |
| **Ever discussed herbal medicine usage with the physician** | 34 | 17(12.3) | 1.8 (0.9, 3.6) | 0.7 (0.2, 2.3) |
| **Intention** | | | | |
| **Intend to use only conventional medicine for the treatment of disease(agree)** | 294 | 70(50.7) | 0.04(0.02, 0.1) | 0.1(0.1, 0.3)*** |
| **Attitude[1]** | | | | |
| Herbal medicines are beneficial | 158 | 113(83.1) | 0.4 (0.2, 0.7) | 3.3(1.5, 7.6)**** |
| Conventional medicines are safe | 329 | 114(82.6) | 0.4 (0.2, 0.8) | 0.7 (0.2, 2.0) |
| The use of both conventional medicines and herbal medicines is safe | 147 | 113(81.9) | 26.6(15, 46.8) | 7.7(3.5, 17.0)*** |
| **Perceived Behavioural Control[1]** | | | | |
| e availability and accessibility of a treatment mode make it an easy choice for caregivers to use | 278 | 129(93.5) | 8.2 (4.0, 16.9) | 2.6 (0.9, 7.6) |
| Caregivers use either herbal or conventional medicines because their surrounding dictates it | 226 | 109(79.0) | 3.7 (2.3, 6.0) | 0.9 (0.4, 2.1) |

****P<0.01

***P<0.05

**P<0.1

*P<0.2 (Prob > chi2 = 0.001, R2 = 0.4985)

[1]Indepedent items that were significant at the bivariate analysis, as shown in Table 5

Given that sickle cell disease is a chronic condition that is associated with various devastating symptoms, the probability that both conventional medicine and herbal medicine are used in the management of the disease is expected to be on the increase. Several studies have indicated this growing phenomenon, especially for chronic conditions [3–6, 23, 24]. Against this background, our findings fit into the global prevalence rates, but most importantly, there may be an increased risk of drug-herbal interactions that may be counterproductive. The results of

the study provided partial support for the Theory of Planned Behaviour, given that the intentions to use both therapies predicted behavior (use of both CM and HM). Findings showed that over two-thirds of the caregivers who intended to use both herbal and conventional used both therapies. The Theory of Planned Behaviour proposes that intention is the most proximal determinant of behavioral outcomes, with attitudes, subjective norms, and perceived behavioral control proposed to predict intention [21]. The Theory of Planned Behaviour has been used successfully by many researchers to predict a variety of behaviors, including the practice of herbal medicine use [25–27].

In this study, findings showed that attitude and perceived behavioral control predicted behavior e.g. caregivers who believed that the use of both herbal and conventional medicine is safe and beneficial were more likely to use both herbal and conventional medicine. The findings could further be elaborated by the attitudes the caregivers have towards the use of conventional medicine only. Several studies have also revealed that belief in the efficacy and safety of herbal medicine, and intention to use herbal medicine or combined therapy seemed to be key determinants of future use of combined therapy compared to those who believed in the efficacy of conventional medicine alone [13, 14, 28].

In sickle cell disease, the low expectancy of a cure for these genetic conditions and a feeling of failure of conventional medicine are possible drivers for the increased concurrent use of herbal and conventional medicine. Relative to that, past experiences with herbal or conventional medicine and the emergence of herbal medicines that assimilate modern medicine substances could also have a contributory role towards the growing concurrent use of herbal and conventional medicines. According to a study done on con-current use of herbal and orthodox medicine in Ghana, it was noted that 25% of the respondents used herbal and conventional medicines owing to the perception that both medicine types work together for the management of the condition and 23.1% believed that the synergic effect of both medicines when combined in treating the prevailing disease condition, works better [28]. This finding corresponds with this study as results have shown a significant association between the use of herbal medicine and conventional medicine and the caregivers who believe that the use of both conventional medicine and herbal medicine is safe. Several studies also revealed that belief in the efficacy and safety of herbal medicine, and intention to use herbal medicine or combined therapy seemed to be key determinants of future use of combined therapy compared to those who believed in the efficacy of conventional medicine alone [13, 14, 28]. However, the combined use of herbal medicine and conventional medicine could hamper the standard management measures of sickle cell disease, by interfering with the potency of conventional medicines, and causing drug toxicity as most herbal medicines are not quantified before use. Nevertheless, a meta-analysis has concluded that attitude towards the behavior is the most important predictor of health behavior intention [27].

We found that perceived behavioral control was significantly associated with the use of combined therapy. More than a third of those who believed that the decision to choose the mode of treatment was under their complete control reported that they used combined therapy, with a slightly higher percentage of use of combined therapy reported among those who believed that the decision to use a given mode of treatment is dictated by its availability and accessibility. Affordability and accessibility are the reasons for using both medicines concurrently [24, 29–31]. This finding suggests that perceived behavioral control may have served as a proxy measure of actual control among the myriad of other influences that can impact the use of herbal and conventional medicine. Profoundly, caregivers are more likely to use both CM and HM on the basis of previous experiences owing to the chronic nature of the disease.

Subjective norms, however, did not emerge as a significant predictor of behavior. Notably, a higher proportion (60.3%) of caregivers who believed that the opinions of their friends

regarding their choice of treatment for sickle cell disease management were important to them reported higher use of combined therapy. The absence of significant findings for the subjective norm in predicting behavioral intention is likely to be related to the limited variability of the construct. Most caregivers are likely to use both CM and HM owing to the pressure from other peer users. For example, in this study, we also noted that caregivers were significantly more likely to hear about the concurrent use of these medicines from friends or fellow caregivers hence, more likely to carry out the practice. In addition, the study showed that caregivers who had knowledge that herbal medicines were used for the treatment of sickle cell disease were significantly more inclined to use both herbal and conventional medicine concurrently for disease management. Therefore, some of the subjective norms and perceived behavioral control's effect on the concurrent use of both therapies were explained by the attitude based on knowledge and past experiences.

Several studies in Nigeria and Norway found an association between the concurrent use of herbal and orthodox medicines and several socio-demographic characteristics including age, sex, level of education, and income level [28, 32, 33]. This study however found age and lower levels of education of respondents as the only factors associated with the concurrent use of herbal and orthodox medicines with individuals aged 60+ almost 12 times more likely to combine the two forms of drugs. This could be attributed to the long-lived experiences of the elderly which provides them with an adequate comparison of events between herbal medicine and conventional medicine use. The elderly caregivers are also less likely to be influenced by medical knowledge on disease management and are more likely to try other remedies or self-medicate with all forms of drugs before hospital visitation. These study findings are in congruence with a study done by [32] who also found that the usage of herbal and orthodox medicines increases with age. However, it is contrary to findings from a study done by Ameade et al, 2018 which showed that the concurrent use of herbal-conventional medicine was common among people below 30 years [28]. The variations could be attributed to the different sample sizes and populations used.

Caregivers with lower education are more inclined to use combination therapy as they could be less privileged income-wise, have less knowledge on the standard of care for sickle cell disease, and are also bound to be easily influenced by friends, relatives, and others into using both remedies or herbal medicine. This study's findings are similar to a recent study done in Sierra Leone which also revealed that the use of herbal medicine is greatly linked to low educational levels [34]. However, this is in contrast with a study done in Nigeria [13] together with other studies done in the USA [35]; which revealed that higher levels of education were significantly associated with more usage of herbal medicine. Other studies, including Oreagba and colleagues, found that, despite the high prevalence (66.8%) of herbal medicine use by their study participants, there was no statistical significance between the level of education and herbal medicine use [23, 29]. Collectively, our findings and findings from previous studies suggest that age, education level, and beliefs associated with a certain remedy greatly contribute to the usage of both herbal and conventional medicines for the management of chronic health conditions [19, 28, 34].

Our study had some limitations as well as strengths. The period of withdrawal from the use of either herbal or conventional medicine or both was not captured is one of the study limitations. Capturing time could have been important in assessing how long a caregiver had withdrawn from a certain therapy. The other limitation is that this study was conducted in a health facility setting and caregivers could have feared to admit that they were using herbal medicines in the setting where they receive conventional medicine to treat children with sickle cell disease. If this happened, then, the prevalence of herbal medicine use alone or combined herbal and conventional medicine could have been underestimated. We tried to minimize this

limitation by reassuring the caregivers of the confidentiality of the interviews as well as conducting interviews away from the hospital setting. It is important to note that we did not collect data about the caregivers' relationship with the child; that is if they were parents, guardians, or other relatives. As such, we were not able to detect if there were any differentials in the administration of herbal or conventional medicine in the treatment of sickle cell disease based on the relationship between the child and the caregiver. This is an aspect that should be explored in future studies.

Besides, since sickle cell disease usually manifests from around six months [36, 37], it would also have been important to enrol caregivers with children starting from six months of age. However, we recruited caregivers with children who were older than six months. It is likely that we could have missed some views from caregivers within this age bracket. However, we intentionally excluded children below one (1) year because during this period, when the disease has just started to manifest, most caregivers may not have conclusive evidence of the disease, as they are still pondering between sickle cell disease or any other disease. Moreover, during this period, the majority of caregivers entirely trust health workers and fully rely on conventional medicine for disease management. However, by the time the child reaches one year of age, caregivers are more likely to look out for other remedies with the realization that the disease is not only a burden but also chronic and incurable. Nevertheless, we believe that our study is unique in that most previous studies have focused mainly on the use of herbal medicine alone or on the use of other complementary medicines in the treatment of sickle cell disease but not on the concurrent use of herbal and conventional medicine. In addition, the study, unlike other studies, was informed by the theory of planned behavior to determine caregivers' behavioral intention to use either herbal or conventional medicine use. Based on the study's findings, the percentage of caregivers using combined therapy to treat children with sickle cell disease is relatively high and this could compromise the health of these children as they are prone to drug toxicities and drug-drug interactions. Furthermore, disease complications could also arise as there are possibilities of herbal medicines interfering with the drug potency of conventional medicines.

## Conclusion

Slightly more than one-third of the caregivers reported the use of combined herbal and conventional medicines to treat children with sickle cell disease, the majority of whom were of older age (>60%) and had lower secondary education. Beliefs in the efficacy and safety of those who believed in the safety and/or efficacy of herbal medicine and beliefs in the safety of using both CM and HM were strongly associated with the use of combined HM and CM. These findings suggest a need for targeted health promotion to educate caregivers about the dangers of using both herbal and conventional medicines in treating children with sickle cell disease in this setting.

## Supporting information

**S1 Dataset. Dataset used in the analysis.**
(XLSX)

## Acknowledgments

We acknowledge the staff of Jinja Regional Referral Hospital for their cooperation and support during the study process. We are also grateful to the research assistants that participated in the

data collection process. Our special regards also go to the study participants who took the time to take part in this study.

## Author Contributions

**Conceptualization:** Consiliate Apolot, Joseph K. B. Matovu.

**Data curation:** Consiliate Apolot.

**Formal analysis:** Consiliate Apolot, Samuel Baker Obakiro, David Mukunya, Peter Olupot-Olupot, Joseph K. B. Matovu.

**Investigation:** Consiliate Apolot.

**Methodology:** Consiliate Apolot.

**Project administration:** Consiliate Apolot.

**Resources:** Consiliate Apolot.

**Software:** Consiliate Apolot.

**Supervision:** Consiliate Apolot, Samuel Baker Obakiro, David Mukunya, Peter Olupot-Olupot, Joseph K. B. Matovu.

**Validation:** Consiliate Apolot, Joseph K. B. Matovu.

**Visualization:** Consiliate Apolot.

**Writing – original draft:** Consiliate Apolot.

**Writing – review & editing:** Consiliate Apolot, Samuel Baker Obakiro, David Mukunya, Peter Olupot-Olupot, Joseph K. B. Matovu.

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
