## [Decision Letter · Decision Letter 0]

23 May 2023

PONE-D-22-34542Caregivers’ Use of Herbal and Conventional Medicine to Treat Children with Sickle Cell Disease at Jinja Regional Referral Hospital, Eastern Uganda: A cross-sectional StudyPLOS ONE

Dear Dr. Apolot,

Thank you for submitting your manuscript to PLOS ONE. After careful consideration, we feel that it has merit but does not fully meet PLOS ONE’s publication criteria as it currently stands. Therefore, we invite you to submit a revised version of the manuscript that addresses the points raised during the review process.

We look forward to receiving your revised manuscript.

Kind regards,

Aloysius Gonzaga Mubuuke

Academic Editor

PLOS ONE

Additional Editor Comments:

The paper is relevant for the journal readership. The authors need to clarify on how sampling was done and also proof-read the paper to strengthen the grammar and language.

Reviewers' comments:

Reviewer's Responses to Questions

**Comments to the Author**

1. Is the manuscript technically sound, and do the data support the conclusions?

Reviewer #1: Partly

Reviewer #2: Yes

2. Has the statistical analysis been performed appropriately and rigorously? 

Reviewer #1: Yes

Reviewer #2: Yes

3. Have the authors made all data underlying the findings in their manuscript fully available?

Reviewer #1: Yes

Reviewer #2: Yes

4. Is the manuscript presented in an intelligible fashion and written in standard English?

Reviewer #1: Yes

Reviewer #2: Yes

5. Review Comments to the Author

Reviewer #1: The manuscript describes a phenomenon that is common in Africa. Some findings were quite interesting. The use of herbal remedies among individuals with tertiary education, over 50% of the cohort used herbal remedies. The demographics was also quite interesting. Most of the respondents were unmarried. It would have been important to distinguish the caregivers that were actual parents in the study population.

Reviewer #2: The manuscript is well written, results are sound and support conclusions made. Methodology is largely logical except for a few things

1. The sampling criteria / procedure is not elucidated. Did you consecutively or randomly sample participants? Please clearly include it in the text.

2. Did you pretest the questionnaire you developed from the literature for suitability in your setting? Please provide more information on this.

3. Usually sickle cell disease manifests from around 6 months, by recruiting caretakers of patients from 1 year old, did you intentionally exclude those of infants receiving treatment? If so, clearly state it with reason.

6. PLOS authors have the option to publish the peer review history of their article (what does this mean?). If published, this will include your full peer review and any attached files.

Reviewer #1: No

Reviewer #2: **Yes: **Andrew Marvin Kanyike

---

## [Author Response · Author response to Decision Letter 0]

11 Jun 2023

REVIEWER ONE 

The manuscript describes a phenomenon that is common in Africa. Some findings were quite interesting. The use of herbal remedies among individuals with tertiary education, over 50% of the cohort used herbal remedies. The demographics were also quite interesting. Most of the respondents were unmarried. It would have been important to distinguish the caregivers that were actual parents in the study population.

RESPONSE: We thank the reviewer for these nice compliments and observations. The suggestion to distinguish caregivers who were actual parents from those who were not is very important as this would show whether there were differentials in terms of herbal/conventional drug administration by the actual parents and those who were not. However, in our study questionnaire, we did not include a question that would distinguish the actual caregiver who was a parent from one who was not and therefore, we are not able to carry out that categorization. We acknowledge this as a limitation and have clearly stated it in the discussion section on page 27. 

REVIEWER TWO 

The manuscript is well written, the results are sound, and support the conclusions made. The methodology is largely logical except for a few things.

1. The sampling criteria/procedure is not elucidated. Did you consecutively or randomly sample participants? Please clearly include it in the text.

RESPONSE: We thank the reviewer for the pleasant observations and comments. Study participants were recruited by consecutive sampling. The sampling criteria have been clearly indicated under the sampling strategy within the materials and methods section. For details, please see pages 6 and 7.

2. Did you pretest the questionnaire you developed from the literature for suitability in your setting? Please provide more information on this.

RESPONSE: Yes, Questionnaires were pretested on 5 caregivers of children with sickle cell disease who were randomly chosen from the sickle cell clinic. More information has been provided under the data collection tools within the methods section. For details, please see page 7.

3. Usually sickle cell disease manifests from around 6 months, by recruiting caretakers of patients from 1 year old, did you intentionally exclude those infants receiving treatment? If so, clearly state it with reason.

RESPONSE: We intentionally excluded children with sickle cell disease of 1 year and below because, at 6 months, when the disease has just started to manifest, most caregivers do not have conclusive findings of the diagnosis of the disease and are most likely to rely on conventional medicine for disease management. However, with time, caregivers are more likely to look out for other remedies with the realization that the disease is chronic, incurable, and burdensome. We have, however, acknowledged this as a limitation in the discussion section. For details, see page 27.

---

## [Editor Report · Decision Letter 1]

14 Jun 2023

PONE-D-22-34542R1Caregivers’ use of herbal and conventional medicine to treat children with sickle cell disease at Jinja Regional Referral Hospital, eastern Uganda: a cross-sectional studyPLOS ONE

Dear Dr. Apolot,

Thank you for submitting your manuscript to PLOS ONE. After careful consideration, we feel that it has merit but does not fully meet PLOS ONE’s publication criteria as it currently stands. Therefore, we invite you to submit a revised version of the manuscript that addresses the points raised during the review process.

We look forward to receiving your revised manuscript.

Kind regards,

Aloysius Gonzaga Mubuuke

Academic Editor

PLOS ONE

Journal Requirements:

Additional Editor Comments (if provided):

Thank you for making the revisions. Some aspects in the paper still need clarity:

1. You should define what you mean by `Care givers` in the context of your study.

2. Tables 4 and 5 should reflect the confidence intervals beyond presenting p-values

3. Regarding results in Table 6, what statistical tests were done with this particular data? There are no significant levels presented and no confidence intervals.

4. The theory of planned behaviour should be explained clearly in your introduction.

5. The discussion should show the strengths of the paper and implications of clinical management of children with sickle cell in relation to the study findings

6. Adult age in Uganda begins at 18 years. How then can you say you included children 1-18 years?
---

## [Author Response · Author response to Decision Letter 1]

2 Aug 2023

EDITOR’S COMMENTS

1. You should define what you mean by `Care givers` in the context of your study.

RESPONSE: We thank the reviewer for the keen comments and observations. 

The definition of caregivers in the context of this study has been defined in the Methods and materials section under study population on page 6.

2. Tables 4 and 5 should reflect the confidence intervals beyond presenting p-values

RESPONSE: For both tables 4 and 5, analysis was re-done and the confidence intervals have been reflected. This can be seen in the Results section on pages 16 for Table 4 and page 17 for Table 5. 

3. Regarding the results in Table 6, what statistical tests were done with this particular data? There are no significant levels presented and no confidence intervals.

RESPONSE: We thank you for the inquiry.

A binary logistic regression was re-run to compute the p-values and the confidence intervals. This has been reflected in Table 6, in the Results section on page 19. The statistical tests done have also been re-emphasized on page 18.

4. The theory of planned behavior should be explained clearly in your introduction.

RESPONSE: We thank you for the observation.

The theory of planned behavior has been clearly explained in the Introduction section on page 4.

5. The discussion should show the strengths of the paper and implications of clinical management of children with sickle cell in relation to the study findings

RESPONSE: Thank you for the observations.

Some of the strengths of the study were already highlighted in the Discussion section, nevertheless; we have added another strength of the paper. This is clearly indicated in the Discussion section on page 27. The implications of clinical management of children with sickle cell in relation to the study’s findings have also been added to the Discussion section on page 27. 

6. Adult age in Uganda begins at 18 years. How then can you say you included children 1-18 years?

RESPONSE: Thank you for the inquiry. 

The study only included children from 1 year to those below 18 years. This has been re-emphasized in the method and materials sections under the sampling procedures and study population. This has been clarified on pages 6 and 7. 

Other changes made in the manuscript

For Table 3, we opted to make some changes as the table does not show associations but rather a comparison of the proportions of caregivers who used a combined therapy against those who did not use based on the selected sickle cell disease management characteristics.

But for the rest of the tables, Tables 4, 5, and 6, the confidence intervals have been reflected beyond the p-values, as suggested by the editor.

---

## [Editor Report · Decision Letter 2]

21 Aug 2023

Caregivers’ use of herbal and conventional medicine to treat children with sickle cell disease at Jinja Regional Referral Hospital, eastern Uganda: a cross-sectional study

PONE-D-22-34542R2

Dear Dr. Apolot,

We’re pleased to inform you that your manuscript has been judged scientifically suitable for publication and will be formally accepted for publication once it meets all outstanding technical requirements.

Kind regards,

Aloysius Gonzaga Mubuuke

Academic Editor

PLOS ONE
---

## [Editor Report · Acceptance letter]

30 Aug 2023

PONE-D-22-34542R2 

Caregivers’ use of herbal and conventional medicine to treat children with sickle cell disease at Jinja Regional Referral Hospital, eastern Uganda: a cross-sectional study 

Dear Dr. Apolot:

I'm pleased to inform you that your manuscript has been deemed suitable for publication in PLOS ONE. Congratulations! Your manuscript is now with our production department. 

Kind regards, 

on behalf of

Dr. Aloysius Gonzaga Mubuuke 

Academic Editor

PLOS ONE